# A Lanosteryl Triterpene (RA-3) Exhibits Antihyperuricemic and Nephroprotective Effects in Rats

**DOI:** 10.3390/molecules25174010

**Published:** 2020-09-02

**Authors:** Nomadlozi Blessings Hlophe, Andrew Rowland Opoku, Foluso Oluwagbemiga Osunsanmi, Trayana Georgieva Djarova-Daniels, Oladipupo Adejumobi Lawal, Rebamang Anthony Mosa

**Affiliations:** 1Department of Biochemistry and Microbiology, University of Zululand, KwaDlangezwa 3886, South Africa; nomadlozih4@gmail.com (N.B.H.); opokuA@unizulu.ac.za (A.R.O.); trayana.djarova@gmail.com (T.G.D.-D.); 2Department of Agriculture, University of Zululand, KwaDlangezwa 3886, South Africa; alafin21@yahoo.com; 3Products Research Unit, Department of Chemistry, Lagos State University, 102101 Lagos, Nigeria; jumobi.lawal@lasu.edu.ng; 4Department of Biochemistry, Genetics and Microbiology, Division of Biochemistry, University of Pretoria, Hatfield 0028, South Africa

**Keywords:** RA-3, nephrotoxicity, triterpene, antioxidant, hyperuricemia

## Abstract

Considering the global health threat posed by kidney disease burden, a search for new nephroprotective drugs from our local flora could prove a powerful strategy to respond to this health threat. In this study we investigated the antihyperuricemic and nephroprotective potential of RA-3, a plant-derived lanosteryl triterpene. The antihyperuricemic and nephroprotective effect of RA-3 was investigated using the adenine and gentamicin induced hyperuricemic and nephrotoxicity rat model. Following the induction of hyperuricemia and nephrotoxicity, the experimental model rats (Sprague Dawley) were orally administered with RA-3 at 50 and 100 mg/kg body weight, respectively, daily for 14 days. Treatment of the experimental rats with RA-3, especially at 100 mg/kg, effectively lowered the serum renal dysfunction (blood urea nitrogen and creatinine) and hyperuricemic (uric acid and xanthine oxidase) biomarkers. These were accompanied by increased antioxidant status with decrease in malondialdehyde content. A much improved histomorphological structure of the kidney tissues was also observed in the triterpene treated groups when compared to the model control group. It is evident that RA-3 possesses the antihyperuricemic and nephroprotective properties, which could be vital for prevention and amelioration of kidney disease.

## 1. Introduction

The recent massive hike of kidney disease on the global rankings of disease burden signals public health crisis [1]. Kidney disease (nephropathy) is a product of multiple factors that inflict damage to nephron, renal parenchyma, and subsequent renal failure, if diagnosis and treatment are delayed [2]. Amongst various risk factors such as diabetes, hypertension, hyperuricemia, and infections, drug-induced nephrotoxicity is considered one of the significant contributors to both acute and chronic kidney disease [3]. This is attributable to increased exposure of the general population to a large number of prescribed and over-the-counter drugs as well as a variety of other ingested foodstuff and environmental intoxicants [4]. Hyperuricemia, a condition characterized by abnormally elevated uric acid levels in the blood, is considered an independent risk factor of kidney disease [5,6]. The underlying kidney damaging effects of uric acid are associated with tubular toxicity, vasoconstriction, oxidative stress, and inflammation [6,7].

Though nephropathy is harmful, with early diagnosis it is treatable to prevent end-stage renal failure. Depending on the underlying cause and stage of the injury, there are various drugs currently in clinical use against hyperuricemia and nephropathy. These drugs include captopril, allopurinol, and some diuretics (loop and thiazide). Despite efficacy of these drugs, they are also associated with some adverse reactions such as hypersensitivity syndrome, acute interstitial nephritis, and electrolyte imbalances [8,9]. The discovery of new nephroprotective drugs with improved safety profile and tolerance are encouraged. Since tissue damages are commonly prone to oxidative stress and excessive inflammatory response [10], new therapeutic drugs with strong antioxidant and anti-inflammatory activities would be preferred.

There is substantial evidence on potential role of various medicinal plants, as crude extracts or pure isolated compounds [2,11,12,13], in protecting the kidney from various insults thus, maintaining its integrity and functions. Even though methyl-3β-hydroxylanosta-9,24-dien-21-oate (RA-3), a lanosteryl triterpene from *Protorhus longifolia* (Bernh) Engl., has displayed significant various bioactivities such as antihyperlipidemic [14], antihyperglycemic [15], and cardioprotective [16] effects, its nephroprotective potential has not been reported. However, the triterpenes from this plant have previously exhibited lack of cytotoxic effects on human embryonic kidney cell lines [17]. To continue exploring potential significant bioactivities of this plant-derived bioactive compound, the current study investigated its antihyperuricemic and nephroprotective potential in adenine-gentamicin induced hyperuricemic and nephrotoxicity rat model. Adenine and gentamicin induced kidney damage is characterized by increased renal tubular cell death and increased blood urea nitrogen (BUN), creatinine (Cr), and uric acid (UA). All these parameters are commonly observed in human clinical subjects of nephropathy.

## 2. Results

### 2.1. Confirmation of the Isolated Lanosteryl Triterepene (RA-3)

RA-3 was routinely isolated and purified from the chloroform extract of *P. longifolia* stem bark using chromatographic techniques. The physical (white crystals, >95% pure, melting point 204–205 °C) and spectral data (IR (KBr) v_max_ = 3469, 1683 cm^–1^, molecular formula C_31_H_50_O_3_) of this compound were in agreement with previous reports from our laboratory [14,18]. The expected chemical structure of the compound was confirmed as methyl-3β-hydroxylanosta-9,24-dien-21-oate (Figure 1).

### 2.2. Body Weight Changes

Body weights of the rats were monitored for the duration of the study. A normal increase in body was observed in the normal. However, the model group showed a slow increase in body when compared to the normal rats. A slight difference in body weight increase was also observed in the RA-3 and allopurinol treated rats (Table 1).

### 2.3. Changes on Serum Levels of Some Renal Dysfunction Biomarkers

Changes in serum levels of some common renal dysfunction biomarkers were analyzed following treatment of the ailing rats with RA-3. Elevated levels of serum creatinine (Cr, 4.8 ± 1.78 mg/dL), angiotensin converting enzyme (ACE, 138.0 ± 39.61 U/L), and blood urea nitrogen (BUN, 22.5 ± 1.11 mg/dL), were observed in the model control group (MC). However, treatment of the rats with RA-3, especially at 100 mg/kg, effectively lowered these renal dysfunction biomarkers (Cr, 0.92 ± 0.23 mg/dL; ACE 75.5 ± 10.9 U/L; BUN 5.75 ± 0.29 mg/dL) near to the values of the normal control group (Figure 2). The results (Cr and BUN) obtained from RA-3 (100 mg/kg) treated groups were also comparable to the group treated with allopurinol.

### 2.4. Serum Levels of Xanthine Oxidase, Uric Acid and Interleukin-6

While higher serum levels of UA (0.61 ± 0.10 mg/dL), xanthine oxidase (XO, 1.48 ± 0.22 mU/mL), and interleukin-6 (IL-6, 0.71 ± 0.28 pg/mL) were observed in the model control group when compared to the normal control, RA-3, in a concentration dependent manner, significantly decreased the serum levels of UA (0.34 ± 0.08; 0.16 ± 0.02 mg/dL, *p* < 0.01) and XO (0.62 ± 0.12; 0.44 ± 0.08 mU/mL, *p* < 0.05). Though the difference was not significant, a relatively lower serum levels of IL-6 were also observed in the RA-3 treated groups when compared to the model control. Similar results were observed in the allopurinol (a known XO inhibitor and hypouricemic drug) treated group (Figure 3) in which a significant decrease in the serum levels of AU and XO were observed.

### 2.5. Serum Biomarkers of Oxidative Stress

Decreased serum levels of superoxide dismutase (SOD) activity, reduced glutathione (GSH) content and thus total antioxidant status along with increased malondialdehyde (MDA) content were observed in the untreated group (model control) when compared to the normal, RA-3, and allopurinol treated groups (Table 2), an indication of oxidative stress. Nevertheless, treatment of the animals with RA-3 at both concentrations (50 and 100 mg/kg) improved the levels of the antioxidants while decreasing the MDA content.

### 2.6. Histological Analysis of the Kidney Tissues

The results of histological analysis are presented in Figure 4. The sections of kidneys from the normal group showed a normal architecture with intact glomerulus and renal tubules. A marked damage to the kidney indicated by robust display of glomerular congestion, dilated renal tubules and epithelial degeneration was observed on the kidney sections from the model control group. However, kidney sections from the rats treated with RA-3, especially at 100 mg/kg showed a much improved histomorphological structure of the kidney (Figure 4v), characterized by minimal renal tubular dilation and necrosis, indicative of recovery.

## 3. Discussion

Considering the global health threat posed by kidney disease burden, a search for new nephroprotective drugs from our local flora could prove a powerful strategy to respond to this health threat. Potential of plant-derived triterpenes as new nephroprotective agents has been demonstrated using various animal models [13,19].

This study investigated the antihyperuricemic and nephroprotective potential of RA-3, a lanosteryl triterpene from *P. longifolia*. The increased serum levels of these biomarkers along with a robust display of glomerular congestion and epithelial degeneration in the kidney samples from the MC group confirmed the induction of hyperuricemia and nephrotoxicity in the rats. It was interesting to observe the significant decrease in serum levels of these biomarkers (UA, BUN, Cr) with improved histomorphology of the kidney tissues from the rats treated with RA-3 at both concentrations. The results served as an indication of the hypouricemic and nephroprotective potential of the triterpene. Substantial experimental evidence has also demonstrated the significant nephroprotective properties of some other plant-derived triterpenes [13,19,20].

While renal toxicity of gentamicin is strongly linked to oxidative stress [3], adenine’s toxicity is associated with renal tubular obstruction and hyperuricemic effect [21]. Hyperuricemia is considered an independent risk factor of kidney disease [5]. Blood UA levels are physiologically regulated by XO activity, an enzyme that catalyzes the formation of UA from xanthine. The XO catalyzed reaction also generates reactive oxygen species (ROS) which can trigger inflammatory reactions and consequent tissue damage [22]. The reduced serum levels of UA along with decreased XO activity in the triterpene treated groups supported the antihyperuricemic effect of RA-3. These observations were also similar to those of the group treated with allopurinol, a standard hyporuricemic drug. Experimental evidence has shown that antihyperuricemic compounds do possess nephroprotective effect [19,23].

Hyperuricemia is known to cause, among others, vascular smooth muscle cell proliferation, endothelial dysfunction, and increased IL-6 synthesis, all of which may contribute to the progression of chronic kidney disease [9]. Though there was no significant difference between the model control and the treated nephrotoxic groups, the relatively lower serum levels of IL-6 in the RA-3 and allopurinol treated groups were observed. This could further be correlated with the suppressed XO activity in these groups. Since XO activity is known to trigger inflammatory reactions and consequent tissue damage [22], the observed results further indicated the tissue protective potential of the triterpene.

It has further been demonstrated that even mild elevations in serum UA can cause hypertension and renal microvascular disease without causing urate crystal deposition in kidneys [24]. The hypertensive effect of elevated serum UA has been associated with activation of the renin-angiotensin system (RAS) [7]. The vasoconstrictive effect of RAS is mediated by ACE, an enzyme that catalyzes conversion of angiotensin I to a bioactive angiotensin II. The observed lower ACE activity in the lanosteryl triterpene treated groups further suggests the potential role of RA-3 in management of hypertension, another important risk factor of nephropathy [25]. The role of ACE inhibitors in prevention of renal function deterioration is well-established [26,27]. The hypouricemic effect exhibited by RA-3 may also be vital in the prevention of other ailments known to be associated with hyperuricemia [28,29,30].

Furthermore, despite the ROS-mediated renal toxicity of gentamicin and adenine, the improved serum total antioxidant status, GSH and SOD along with reduced MDA content in the rats treated with RA-3 indicated its tissue protective potential against oxidative damage. It is interesting to note from these results that the ability of RA-3 to enhance endogenous antioxidant defense while inhibiting lipid peroxidation has previously been demonstrated in streptozotocin (STZ)-induced diabetes and isoproterenol (ISO)-induced myocardial injury in rat models [15,16]. It is also worth noting that similar to gentamicin induced renal tubular toxicity, the necrotic toxicity of STZ and ISO are also associated with oxidative stress [31,32]. Since inflammation and oxidative tissue damage are common culprits in various pathophysiologies including nephropathy, nephroprotective drugs with both anti-inflammatory and antioxidant properties are highly desired. The nephroprotective potential of most medicinal plant extracts and their derived compounds has been associated with their antioxidant and anti-inflammatory activities [11,33].

## 4. Materials and Methods

### 4.1. Reagents

Glutathione assay kit (catalog number: CS0260), xanthine oxidase assay kit (catalog number: MAK078), superoxide dismutase assay kit (catalog number: 19160), antioxidant assay status kit (catalog number: CS0790), malondialdehyde assay kit (catalog number: MAK085), rat interleukin-6 assay kit (catalog number: RAB0333). All the commercial assay kits used were purchased from Sigma Aldrich (St. Louis, MO, USA) 

### 4.2. Plant Extraction and Isolation of the Lanosteryl Triterpene (RA-3)

Plant extract was prepared from stem bark of *P. longifolia* collected from KwaHlabisa, KwaZulu-Natal, South Africa. The plant material (voucher specimen number RA01UZ) was identified and confirmed by Dr. NR Ntuli from Botany Department, University of Zululand. The targeted compound (RA-3) was routinely isolated and purified from the chloroform extract using the method well established in our laboratory [18]. The lanosteryl triterpene was isolated over silica gel chromatography and the pure compound was recrystallized in 100% ethyl acetate. Melting point and spectroscopic data analysis were used, in comparison with literature data [14,18], to confirm chemical structure of the isolated pure (>95%) lanosteryl triterpene.

### 4.3. Animals

The University of Zululand Research Ethics Committee (UZREC) granted ethical clearance (UZREC 171110-030 PGM 2016/321) for use of laboratory animals (Sprague Dawley rats). Sprague Dawley rats (*n* = 30) were collected from the animal unit of the Department of Biochemistry, University of Zululand. The animals were housed in standard cages (maximum of four rats per cage) and maintained at room temperature (23–25 °C; relative humidity ~50%) with a 12:12-h light/dark cycle as per stipulated national guidelines for the care and use of animals. All the rats were in good state of health and were allowed to acclimatize to the experimental set-up for a week before the experiment was conducted. Unless stated otherwise, all the animals had free access to drinking water and normal rat feed throughout the experimental period.

### 4.4. Preparation of RA-3 Solution

Fresh working solutions of RA-3 (50 and 100 mg/kg body weight) were prepared in 2% Tween 20. These concentrations were based on our previous studies on the compound [14,15,16].

### 4.5. Investigation of the Antihyperuricemic and Nephroprotective Potential of RA-3

The adenine and gentamicin induced hyperuricemia and nephrotoxicity rat model, slightly modified from that described by Meng et al. [34] was used to evaluate the antihyperuricemic and nephroprotective effect of the triterpene (RA-3). Sprague Dawley rats of either sex (150–200 g) were randomly divided into two major groups; normal and model groups. The rats in the model group were orally administered with adenine (150 mg/kg body weight) and intraperitoneally injected with gentamicin (40 mg/kg body weight) daily for fourteen days to induce hyperuricemia and nephrotoxicity.

Following induction of nephrotoxicity, the animals in the model group were randomly divided into four groups (II-V) of at least five rats per group (*n* = 5). The normal control (I) and model control (II) groups received drinking water and 2% Tween 20 (vehicle, p.o), respectively, throughout the experimental period. While the positive control group (III) received allopurinol (10 mg/kg, body weight, p.o), the experimental groups IV and V were orally administered with RA-3 at 50 and 100 mg/kg, respectively. The respective drugs were administered daily for a further fourteen days period. All drug administrations were performed between 8:00 and 9:00 AM and the animals were treated in the same order throughout the experimental period. At the end of the experimental period, the rats were fasted for eight hours and euthanized under anesthesia (Pentobarbital at 30 mg/kg body weight, ip). A complete loss of sensation was confirmed by pedal withdrawal reflex before any procedure could be conducted on the rats. The rats were quickly dissected, blood and kidney samples were immediately collected. The collected tissues samples were used for estimation of some biochemical parameters and histopathological indications, respectively.

#### 4.5.1. Biochemical Analysis

The collected blood samples were separately allowed to clot and centrifuged at 1200 rpm for 10 min. Serum was collected and used for estimation of some selected biochemical parameters. The serum levels of total antioxidant status, GSH, SOD, XO, and MDA were estimated using commercial activity assay kits (Sigma Aldrich, St. Louis, MO, USA). The total antioxidant status assay kit non-specifically measures the total concentration of the antioxidant molecules (small molecules and proteins) in a biological sample (serum, cell and tissue lysate etc.). The serum IL-6 content was measured with ELISA kit (Sigma Aldrich, St. Louis, MO, USA). Standard laboratory procedures (Global Clinical & Viral Laboratory, Kwazulu Natal, South Africa) were followed to estimate the serum levels of Cr, UA, ACE, and BUN.

#### 4.5.2. Histological Analysis of Kidney Tissues

Kidney tissues, preserved in 10% neutral buffered formalin, were embedded in paraffin and sectioned according to standard procedures. The sectioned tissues were subjected to Hematoxylin and Eosin staining for routine histopathological examination (Department of Physiology Department, University of KwaZulu-Natal, Kwazulu Natal, South Africa).

### 4.6. Data Analysis

Data were expressed as mean ± standard deviation (SD) of the mean. The results were analyzed by Kruskal-Wallis test followed by Dunn’s post hoc multiple comparison test using GraphPad Prism software (version 6). The statistical differences were considered significant at *p* < 0.05.

## 5. Conclusions

The results of the current study have indicated and confirmed the antihyperuricemic and nephroprotective potential of RA-3. The ACE inhibitory effect and hypouricemic potential displayed by the triterpene are not only crucial in renal protection, but in other related complications such as cardiovascular disease, hypertension and gout arthritis as well. Investigation of the molecular basis of its nephroprotective potential will provide an insight into the triterpene’s therapeutic mechanism.

## Figures and Tables

**Figure 1 molecules-25-04010-f001:**
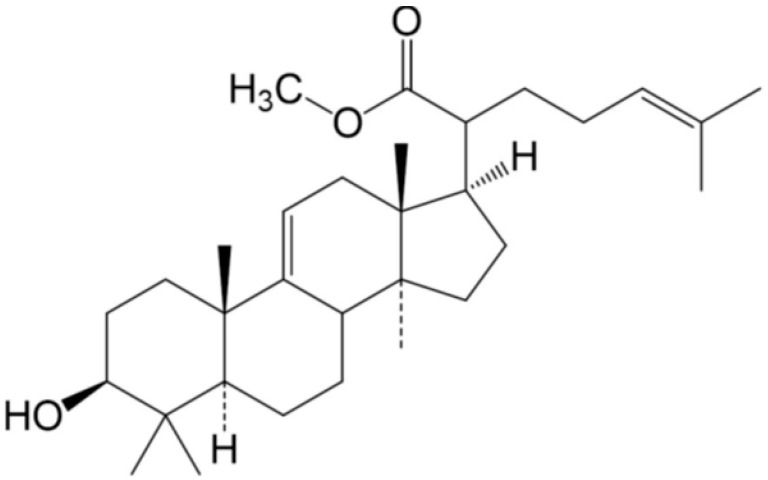
Methyl-3β-hydroxylanosta-9,24-dien-21-oate (RA-3), a lanosteryl triterpene from *P. longifolia*.

**Figure 2 molecules-25-04010-f002:**
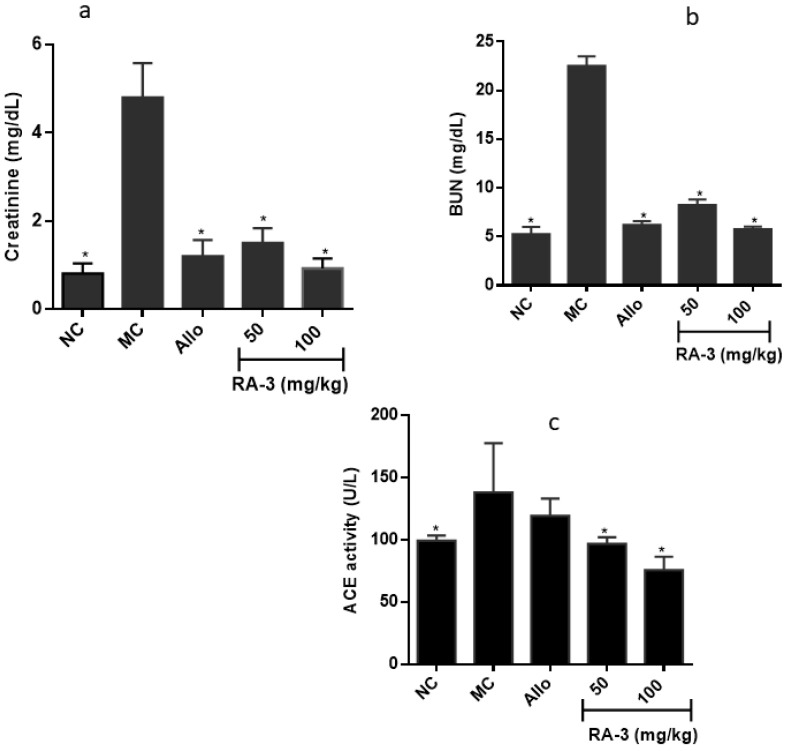
Effect of RA-3 on serum levels of creatinine (**a**), blood urea nitrogen (BUN) (**b**), and angiotensin converting enzyme (ACE) (**c**). Data were expressed as mean ± SD (*n* = 5), * *p* < 0.05 vs. MC group. NC—normal control, MC—model control, Allo—allopurinol.

**Figure 3 molecules-25-04010-f003:**
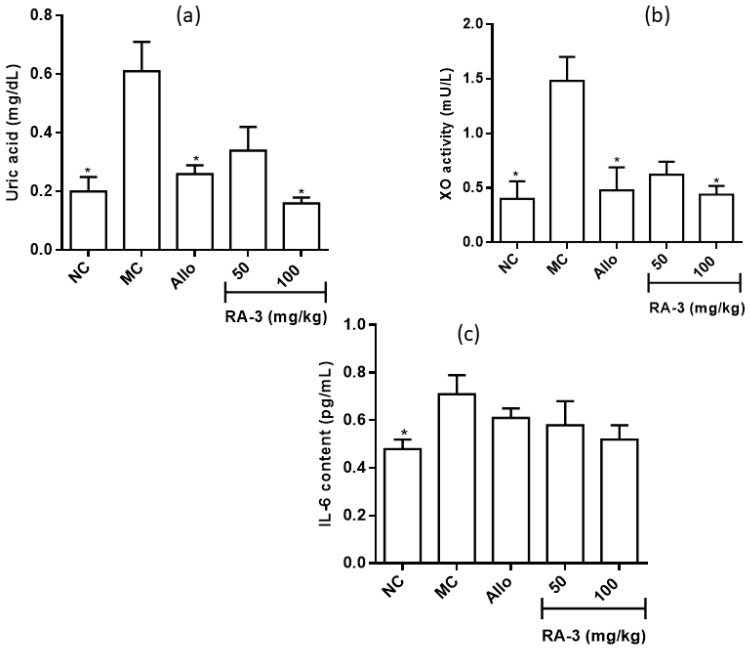
Effect of RA-3 on the serum levels of uric acid (**a**), xanthine oxidase (XO) (**b**) and interleukin-6 (IL-6) (**c**). Data were expressed as mean ± SD (*n* = 5), * *p* < 0.05 vs. MC group. NC—normal control, MC—model control, Allo—allopurinol.

**Figure 4 molecules-25-04010-f004:**
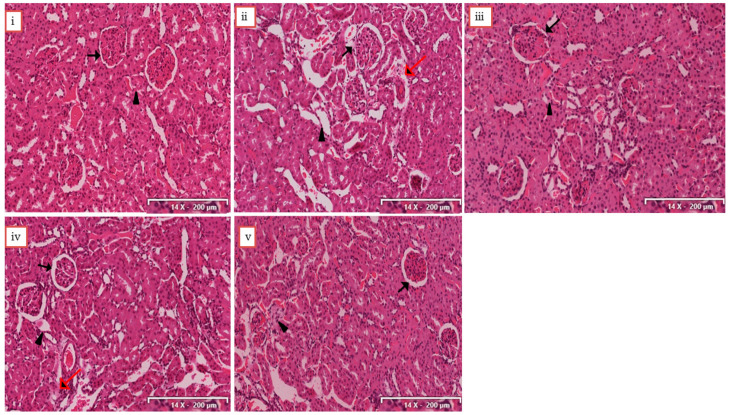
Photomicrographs of the kidney sections of the rats. (**i**) Section of kidney from normal group, showing normal kidney architecture with intact glomerulus (black arrow) and renal tubules (black triangle); (**ii**) section from model control group, showing glomerular congestion, dilated renal tubules, and epithelial degeneration (red arrow), (**iii**) section from group treated with allopurinol, showing minimal improved renal architecture; (**iv**) section from group treated with RA-3 (50 mg/kg), showing regeneration of epithelium, (**v**) section from group treated with RA-3 (100 mg/kg), showing improved renal architecture characterized by epithelial regeneration and reduced glomerular congestion. The indicator size for each image is 14X-200 µm (H&E).

**Table 1 molecules-25-04010-t001:** Body weight changes of the rats after 28 days of the experiment.

Group	Initial Body Weight (g)	Final Body Weight (g)	Body Weight Changes (g)
Normal	192.3 ± 9.33	215.0 ± 4.71	22.7 *
Model control	165.8 ± 14.06	176.9 ± 10.32	11.1
Allopurinol (10 mg/kg)	155.2 ± 21.68	173.1 ± 7.77	17.9 *
RA-3 (50 mg/kg)	179.1 ± 6.20	193.7 ± 15.49	14.6
RA-3 (100 mg/kg)	159.6 ± 10.18	176.8 ± 20.17	17.2

Data were expressed as mean ± SD (*n* = 5). * *p* < 0.05 vs. model control group.

**Table 2 molecules-25-04010-t002:** Effect of RA-3 on the serum levels of oxidative stress markers in the rats.

Group	GSH (nmol/mL)	SOD (Units/mL)	Total Antioxidant Status (mM)	MDA (nmol/µL)
Normal	0.91 ± 0.31 *	71 ± 0.14 *	1.94 ± 0.17 *	0.17 ± 0.06 *
Model control	0.22 ± 0.06	14.3 ± 0.06	0.34 ± 0.02	1.04 ± 0.03
Allopurinol (10 mg/kg)	1.20 ± 0.68 *	21.0 ± 0.03	1.00 ± 0.55	0.20 ± 0. 44 *
RA-3 (50 mg/kg)	0.74 ± 0.22	24. 7± 0.15 *	1.40 ± 0.40 *	0.16 ± 0.04 *
RA-3 (100 mg/kg)	0.80 ± 0.18 *	55.0 ± 0.17 *	1.87 ± 0.20 *	0.14 ± 0.02 *

Data were expressed as mean ± SD (*n* = 5). * *p* < 0.05 vs. model control group.

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
