# Peer review of "A Lanosteryl Triterpene (RA-3) Exhibits Antihyperuricemic and Nephroprotective Effects in Rats"

_molecules, 2020, doi:10.3390/molecules25174010_

Round 1

Reviewer 1 Report

This study investigated the antihyperuricemic and nephroprotective potential of RA-3, a lanosteryl triterpene from P. longifolia. The hypothesis of the study is relevant and well justified. However, I have some comments on the statistical evaluation of the results. First, description of statistical analysis is not detailed and it is hard for readers to understand. Second, normality of data should be confirmed before running ANOVA test. The sample size is very small, it is impossible to evaluate data distribution accurately. I believe that ANOVA test is not appropriate in this case. Finally, all factors included in the analysis should be listed in the description. It is difficult to evaluate if conclusion in this work is relevant before appropriate statistical analyses are conducted.
Minor comment.
The number of animals (total and per treatment) should be described in Material and methods, Animals.

Reviewer 2 Report

Review of the article titled:

A Lanosteryl Triterpene (RA-3) Exhibits Antihyperuricemic and Nephroprotective Effects in Rats

By Nomadlozi et al.

Minor concerns:

Line 52:  Since tissue damage are commonly “subject to”
Can be changed to  "prone to"

Line 55: There is a substantial evidence

I think it can just be “There is substantial evidence” please remove “a”

Major concerns:

I think Kruskal-Wallis statistical test followed by Dunn’s posthoc analysis should be the test of choice. Given the low “N” I don’t think a normality test is appropriate. In that case, non-parametric tests such as Kruskal-Wallis must be used.

  1. Please list all the Catalog numbers and the vendor names of the commercial assays that were used in the study.

  1. Please clarify what is a “total antioxidant assay”. I am not familiar with this assay. A brief explanation of the principle of the assay in the methods section will be helpful.

  1. Was the bodyweight of the rats monitored for the duration of the treatments? Please provide a table showing the body weights of rats.

Reviewer 3 Report

Hlophe N.B et al., were evaluated the antihyperuricemic and nephroprotective effects of a plant-derived Lanosteryl Triterpene (RA-3). Studies reveal that the oral administration of RA-3 to hyperuricemic and nephrotoxicity rat model for fourteen days lowered the serum creatinine, BUN and ACE levels. Further, authors have observed that the treatment significantly decreased the uric acid, xanthine oxidase and improved the serum oxidative stress biomarkers. The manuscript is well-presented, and the design of the study is appropriate enough to achieve the aim mentioned in the manuscript. Overall, results of the manuscript may be beneficial to researchers especially in renal diseases and natural products.

Some inadequacies and suggestions listed below:

  1. I am surprised to see such a high levels of serum creatinine levels in normal controls (8-9mg/dl?), which is more than BUN. In this case, the ratio of BUN to creatinine is important. Authors should quantify the data and mention the explanation.
  2. Authors are suggested to mention the scale bars in all panels of figure 4. For reader convenience, change the colors of orange and blue arrows to a noticeable color.
  3. It is suggested to avoid the repeated information in the article (ex: line 64-66 and 144)
  4. Line 65: It is advised to reframe the sentence ‘…. characterized by increased renal tubular cell death and increased blood urea nitrogen…’

Round 2

Reviewer 1 Report

The revision is satisfactory.

Reviewer 3 Report

Authors have responded to all my questions and made necessary changes to the manuscript. I have no further comments.